# Papaverinol-*N*-Oxide: A Microbial Biotransformation Product of Papaverine with Potential Antidiabetic and Antiobesity Activity Unveiled with In Silico Screening

**DOI:** 10.3390/molecules28041583

**Published:** 2023-02-07

**Authors:** Duaa Eliwa, Amal Kabbash, Mona El-Aasr, Haytham O. Tawfik, Gaber El-Saber Batiha, Mohamed H. Mahmoud, Michel De Waard, Wagdy M. Eldehna, Abdel-Rahim S. Ibrahim

**Affiliations:** 1Department of Pharmacognosy, Faculty of Pharmacy, Tanta University, Tanta 31527, Egypt; 2Department of Pharmaceutical Chemistry, Faculty of Pharmacy, Tanta University, Tanta 31527, Egypt; 3Department of Pharmacology and Therapeutics, Faculty of Veterinary Medicine, Damanhour University, Damanhour 22511, Egypt; 4Department of Biochemistry, College of Science, King Saud University, Riyadh P.O. Box 2455, Saudi Arabia; 5Smartox Biotechnology, 6 Rue Des Platanes, F-38120 Saint-Egrève, France; 6L’institut du Thorax, INSERM, CNRS, UNIV NANTES, F-44007 Nantes, France; 7LabEx Ion Channels, Science & Therapeutics, Université de Nice Sophia-Antipolis, F-06560 Valbonne, France; 8Department of Pharmaceutical Chemistry, Faculty of Pharmacy, Kafrelsheikh University, Kafrelsheikh 33516, Egypt; 9School of Biotechnology, Badr University in Cairo, Badr City 11829, Egypt

**Keywords:** biotransformation, papaverine, protein tyrosine phosphatase 1B, α-glucosidase, pancreatic lipase, antidiabetes, ADMET, antiobesity, molecular docking and dynamics

## Abstract

Bioconversion of biosynthetic heterocyclic compounds has been utilized to produce new semisynthetic pharmaceuticals and study the metabolites of bioactive drugs used systemically. In this investigation, the biotransformation of natural heterocyclic alkaloid papaverine via filamentous fungi was explored. Molecular docking simulations, using protein tyrosine phosphatase 1B (PTP1B), α-glucosidase and pancreatic lipase (PL) as target enzymes, were performed to investigate the antidiabetic potential of papaverine and its metabolites in silico. The metabolites were isolated from biotransformation of papaverine with *Cunninghamella elegans* NRRL 2310, *Rhodotorula rubra* NRRL y1592, *Penicillium chrysogeneum* ATCC 10002 and *Cunninghamella blackesleeana* NRRL 1369 via reduction, demethylation, *N*-oxidation, oxidation and hydroxylation reactions. Seven metabolites were isolated: namely, 3,4-dihydropapaverine (metabolite **1**), papaveroline (metabolite **2**), 7-demethyl papaverine (metabolite **3**), 6,4′-didemethyl papaverine (metabolite **4**), papaverine-3-ol (metabolite **5**), papaverinol (metabolite **6**) and papaverinol N-oxide (metabolite **7**). The structural elucidation of the metabolites was investigated with 1D and 2D NMR and mass spectroscopy (EI and ESI). The molecular docking studies showed that metabolite **7** exhibited better binding interactions with the target enzymes PTP1B, α-glucosidase and PL than did papaverine. Furthermore, papaverinol-*N*-oxide (**7**) also displayed inhibition of α-glucosidase and lipase enzymes comparable to that of their ligands (acarbose and orlistat, respectively), as unveiled with an in silico ADMET profile, molecular docking and molecular dynamics studies. In conclusion, this study provides evidence for enhanced inhibition of PTP1B, α-glucosidase and PL via some papaverine fungal transformation products and, therefore, potentially better antidiabetic and antiobesity effects than those of papaverine and other known therapeutic agents.

## 1. Introduction

The process of bioconversion involves transformation of both natural and synthetic compounds, most frequently into closely related derivatives with lower persistence and toxicity. Such a process is performed with many biological systems, including bacteria, fungi and their enzymes [1,2,3]. Alkaloids represent a class of natural plant products with biological activity that affects mankind, ranging from valuable therapeutic activity to severe toxicity [4,5]. Alkaloids are subjected to microbial transformation to obtain new or improved therapeutic agents [6]. Six metabolites were previously isolated from microbial transformation of papaverine via *Aspergillus niger* NRRL 322, *Beauveria bassiana* NRRL 22864, *Cunninghamella echinulate* ATCC 18968 and *Cunninghamella echinulate* ATCC 1382 [7]. Using a cell culture of *Silene alba*, papaverine was transformed to papaveraldine and 4′ and 6 monodemethyl papaverine [8,9]. Papaveraldine was found to be transformed into *S*-papaverinol and S-papaverinol N-oxide via the fungus *Mucor ramannianus* 1839 [10]. Furthermore, rat bile converted papaverine into four metabolites: 4′-demethyl, 7-demethyl, 6-demethyl and 4′,6-demethyl papaverine [11]. Due to the broad range of biological activities displayed by papaverine, it was subjected to further microbial transformation study through use of a new battery of microorganisms. The present investigation unveiled seven unreported metabolites of papaverine mainly through oxidation reactions. However, 3,4-dihydropapaverine was also isolated. Two of the metabolites were similar to those obtained from incubation of papaverine with rat bile, namely 7-demethyl and 4′,6-demethyl papaverine, which reinforces the presence of parallels between microbial and mammalian metabolic profiles. Papaverine is a selective e inhibitor of antidiabetic target enzyme human protein tyrosine phosphatase 1B (h-PTP 1B) [12,13,14]. A-Glycosidase inhibitors, which have been used to treat type 2 diabetes, could inhibit breakdown of carbohydrates into glucose [15]. Therefore, the antidiabetic potential of papaverine and its metabolites through α-glucosidase inhibition was also investigated in silico for the first time. Obesity is a significant environmental predisposing factor that has a role in development of type 2 diabetes (T2D) as well as its complications [16]. Papaverine is a selective inhibitor of antiobesity target enzyme pancreatic lipase (PL), and it could be used as a valuable new frontier for development of more effective and specific PL inhibitors as potential antiobesity medications [17]. The metabolites of papaverine in this study were also subjected to molecular docking simulations to analyze the binding patterns to h-PTP 1B, PL and α-glucosidase enzymes using MOE software. The microbial transformation of therapeutic agents can reduce delivery times and circumvent speculative chemical synthesis [18,19].

## 2. Results and Discussion

### 2.1. Identification of the Biotransformation Products

Fifty different types of fungi of different classes were used in the initial screening of isoquinoline alkaloid substrate papaverine. Seven biotransformation products of papaverine were isolated using different fungi: *Cunninghamella elegans* NRRL 2310, *Penicillium chrysogeneum* ATCC 10002, *Rhodotorula rubra* NRRL y1592 and *Cunninghamella blackesleeana* NRRL 1369. Structural elucidation of metabolites was based on their mass and NMR spectroscopic data.

The mass spectrum of **1** (SI: S5) displayed a molecular ion peak at *m*/*z* 341, corresponding to a molecular formula of C_20_H_23_NO_4_, which is 2 Da higher than that of papaverine. Peaks at *m*/*z* 326, 310, 309 and 295 corresponded to (M-CH_3_)^+^, (M-OCH_3_)^+^, (M-H-OCH_3_)^+^ and (M-OCH_3_-CH_3_)^+^ fragments, respectively. When compared with papaverine, ^1^H NMR (Table 1, SI:S1) displayed the absence of doublets of one-proton intensity each at δ_H_ 8.33 ppm (J = 5.6 Hz) and 7.53 ppm (J = 5.6 Hz), belonging to two protons, H-3 and H-4, respectively, that are present in papaverine. It also displayed the appearance of two other triplets of two protons’ intensity at δ_H_ 3.47 (J = 7.2 Hz) ppm and 3.07 (J = 7.2 Hz) ppm, assigned to four protons on saturated carbons at H-3 and H-4, respectively, which suggested a reduction in C-3 and C-4 positions. A total of twenty-three protons resonating between δ_H_ 3.07 and 7.38 ppm compared to twenty-one protons in papaverine confirmed reduction in the dihydro derivative. In **1**, ^13^C NMR (Table 2, SI: S2) showed twenty signals corresponding to twenty different carbons and the disappearance of two carbon signals, at δ_C_ 119.6 and 139.0 ppm, present in papaverine. Furthermore, two high field resonances, at δ_C_ 46.5 ppm (C-3) and 26.2 ppm (C-4), substantiated the reduction of papaverine. Other ^13^C NMR data exhibited resemblance to those of papaverine. Carbon multiplicity experiments (DEPT-135 and APT spectra) on metabolite **1** (SI: S3, SI: S4) proved the existence of three “methylene” carbons (attributed to C-α, C-3 and C-4), four “methyl” carbons (C-3′, C-4′, C-7, C-6) and five “methine” carbons (C-5, C-8, C-6′, C-5′, C-2′). The IR data (SI: S6) showed a characteristic broad band of N–H stretching at 3430 cm^−1^, one of aliphatic C-C stretching at 1603 cm^−1^ and one of tertiary alcoholic C–O stretching at 1157 cm^−1^. All information matched that which had been reported for 3, 4-dihydropapaverine [20] (Figure 1).

The mass spectrum of **2** (SI: S11) exhibited a molecular ion peak at **m*/*z** 283, corresponding to a molecular formula of C_16_H_13_NO_4_, which is 56 Da lower than that of papaverine. The ^1^H NMR data of **2** (SI: S7) compared to papaverine (Table 1) showed the disappearance of three sharp proton singlets of 7, 6, 3′ and 4′ methoxy moieties, resonating at δ_H_ 4.02, 3.93, 3.78 and 3.80 ppm, respectively, and the appearance of another four broad singlets that disappeared upon addition of D_2_O, integrating for four protons at δ_H_ 5.06, 5.15, 5.33 and 5.64 ppm, due to four OH groups resulting from the demethylation reaction at C-6, C-7, C-3′ and C-4′, respectively. Of **2**, ^13^C NMR (Table 2 and SI: S8) showed sixteen signals, which corresponded to sixteen different carbon atoms reminiscent to those of papaverine, with the sole difference being the lack of four methoxy carbon signals of papaverine at δ_C_ 55.9, 56.12, 56.10 and 56.4 ppm. Carbon multiplicity experiments (DEPT-135 and APT) on **2** (SI: S9, SI: S10) confirmed the presence of one “methylene” carbon (C- α) and seven “methine” carbons (C-3, C-4, C-5, C-8, C-6′, C-5′ and C-2′). The IR spectrum (SI: S12) of **2** exhibited a hydroxy group band at 3455 cm^−1^, which was missing in the papaverine spectrum as reported (SI: S12). All data were coincident with those published for papaveroline [21] (Figure 1).

The mass spectrum of **3** (SI: S19) exhibited a molecular ion peak at *m*/*z* 325, corresponding to a molecular formula of C_19_H_19_NO_4_, which is 14 Da lower than that of papaverine. Peaks at *m*/*z* 324, 310, 294, 293 and 279 corresponded to (M-H)^+^, (M-CH_3_)^+^, (M-OCH_3_)^+^, (M-H-OCH_3_)^+^ and (M-OCH_3_-CH_3_)^+^ fragments, respectively. The presence of fragments at *m*/*z* 188 and 151 increased the evidence of a C-7 demethylation reaction, as previously reported. The IR spectrum (SI: S20) of **3** showed an OH group band at 3438 cm^−1^, which was missing in the papaverine spectrum. The ^1^H NMR spectra of 3 (Table 1, SI: S13) lacked a singlet signal at δ_H_ 3.93 ppm, integrating for three protons (C-7 methyl group) present in papaverine, and the appearance of another broad singlet, integrating for one proton at δ_H_ 6.02 ppm, due to the OH group resulting from the demethylation reaction. For ^13^C NMR (Table 2, SI: S14), carbon multiplicity experiments (DEPT-135 and APT spectra) (SI: S15 and SI: S16) also showed the disappearance of a signal at δ_C_ 56.10 ppm due to the C-7 methoxy group in papaverine. This was substantiated through extensive analysis of the HMBC and HSQC spectra of **3** (SI: S17, SI: S18). Based on the aforementioned spectral data, which were consistent with published data, metabolite **3** was identified as 7-demethyl papaverine [11] (Figure 1).

The mass spectrum (SI: S27) of **4** displayed a molecular ion peak at *m*/*z* 311, corresponding to a molecular formula of C_18_H_17_NO_4_, which is 28 Da lower than that of papaverine. Peaks at *m*/*z* 310, 296, 280, 279 and 265 corresponded to (M-H)^+^, (M-CH_3_)^+^, (M-OCH_3_)^+^, (M-H-OCH_3_)^+^ and (M-OCH_3_-CH_3_)^+^ fragments, respectively. The presence of fragments at *m*/*z* 188 and 137 increased the evidence of C-6 and C-4′ demethylation reactions, as previously reported [5]. In addition, **4**, when compared to the papaverine, displayed a lack of 6 and 4`-methoxy groups resonating at δ_H_ 4.02 and 3.80 ppm in ^1^H NMR (Table 1, SI: S21). For ^13^C NMR, carbon multiplicity experiments (DEPT-135 and APT spectra) on **4** (Table 2, SI: S22, SI: S23 and SI: S24) showed two methoxy signals at δ_C_ 55.3 and 55.5 ppm, which are less than that of papaverine by two methoxy groups. This was substantiated with extensive analysis of the HMBC and HSQC spectra of **4** (SI: S25 and SI: S26). The IR spectrum (SI: S28) also revealed the presence of a hydroxy band at 3450 cm^−1^, which is absent in the papaverine spectrum, as previously published. Hence, **4** was identified as 6, 4′-didemethyl papaverine [11] (Figure 1).

The ^1^H NMR spectrum of metabolite **5** compared to that of papaverine (Table 1, SI: S29) displayed a lack of doublets of one-proton intensity each at δ_H_ 8.33 (J = 5.6 Hz) ppm and 7.53 (J = 5.6 Hz), belonging to two protons on H-3 and H-4, respectively, that are present in papaverine. It also displayed the appearance of another two singlets of one-proton intensity each, at δ_H_ 6.99 and 2.36 ppm, which belonged to one proton on C-4 and the OH group on C-3, respectively. The later signal disappeared on deuteration, which suggested hydroxylation of C-3. Examination of the ^13^C NMR of **5** (Table 2, SI: S30) showed twenty peaks, which were related to twenty diverse carbon atoms. There was a considerable deshielding of C-3 carbon signals (by 25 ppm) and a considerable shielding of C-4 (by 20 ppm). Other ^13^C NMR data revealed resemblance to those of papaverine. The DEPT-135 spectra of **5** (SI: S31) confirmed the presence of one methylene carbon (assigned to C-α), four methyl carbons (C-3′, C-4′, C-7, C-6) and six methine carbons (C-5, C-8, C-6′, C-5′, C-2′, C-4). The mass spectrum (SI: S32) showed the presence of a molecular ion peak at *m*/*z* 355, presumably relating to a molecular formula of C_20_H_21_NO_5_, which is 16 Da higher than that of papaverine. The IR spectrum (SI: S33) showed an OH band at 3432 cm^−1^, which is not present in the papaverine spectrum, as previously reported. Peaks at *m*/*z* 354, 340, 325 and 309 corresponded to (M-H)^+^, (M-CH_3_)^+^, (M-OCH_3_)^+^ and (M-OCH_3_-CH_3_)^+^ fragments, respectively. Based on these data, metabolite **5** was elucidated as papaverine 3-ol [22] (Figure 1).

The positive ESI-MS spectrum (SI: S38) of **6** exhibited [M+H]^+^ at *m*/*z* 356 of a pseudomolecular ion peak, apparently indicating a molecular formula of C_20_H_22_O_5_N, which is 16 Da higher than that of papaverine (one extra oxygen atom). The ^1^H NMR data of **6** (Table 1, SI: S34) showed that the metabolite spectrum exhibited four sharp three-proton singlets at δ_H_ 3.95, 3.71, 3.76 and 3.81 ppm, assigned to the 6, 7, 3′ and 4′ methoxy groups, respectively. It also lacked a singlet signal of two protons’ intensity at δ_H_ 4.56 ppm, belonging to two protons on carbon α, which is present in papaverine, and the presence of a broad singlet at δ_H_ 6.08 ppm, integrating for one proton. The remaining chemical shifts of proton signals were very close to those of papaverine, as reported. The ^13^C NMR of **6** (Table 2, SI: S35) showed twenty peaks, which related to twenty various carbon atoms similar to the total number of carbon atoms of papaverine, but there was considerable deshielding of C-α (by 30 ppm) and C-1′ (by 3 ppm). Both the DEPT-135 and APT spectra of **6** (SI: S36, SI: S37) exhibited the absence of one methylene carbon (attributed to C-α) present in papaverine and the appearance of a new methine carbon at δ_C_ 72.5 ppm. In addition, seven other methine carbons (C-3, C-4, C-5, C-8, C-2′, C-5′, C-6′,) and four methyl carbons (C-3′, C-4′, C-6, C-7) were observed. The IR spectrum (SI: S39) showed a strong absorption band at 3345 cm^−1^, suggesting the presence of hydroxy functionality, which is lacking in the papaverine spectrum, as formerly reported. Based on **6**’s spectral data, it was identified as papaverinol that was previously isolated from microbial transformation of papaveraldine via Mucor ramannianus 1839 [10], However, this is the first report of papaverinol being isolated from biotransformation of papaverine (Figure 1).

The (+) ESI-MS spectrum (SI: S44) of **7** displayed a pseudomolecular ion peak [M + H]^+^ at *m*/*z* 372, indicating a molecular formula of C_20_H_22_O_6_N, which is 32 Da higher than that of papaverine (two additional oxygen atoms). The ^1^H NMR data of **7** (Table 1, SI: S40) displayed a lack of a singlet of two protons’ intensity at δ_H_ 4.56 ppm belonging to two protons on carbon α, which is present in papaverine, and the appearance of a broad singlet at δ_H_ 6.50 ppm, integrating for one proton, which disappeared upon addition of D_2_O, in addition to a singlet signal at δ_H_ 6.1 ppm due to C-α. There was a slight deshielding in doublets of one-proton intensity at δ_H_ 8.41 (J = 5.2 Hz) ppm and δ_H_ 7.51 ppm (J = 5.2 Hz), belonging to two protons on unsaturated carbons and assigned to H-3 and H-4, respectively. Signals observed at δ_H_ 7.06 (1 H, s) ppm and δ_H_ 7.12 (1 H, s) were assigned to H-5 and H-8, respectively. Examination of the ^13^C NMR of **7** (Table 2, SI: S41) showed twenty peaks, which corresponded to twenty various carbon atoms, similar to the total number of carbon atoms of papaverine, but there was considerable deshielding of C-α (by 30 ppm), C-1′ (by 4 ppm), C-1 (by 3 ppm) and C-3 (by 2 ppm). In carbon multiplicity NMR experiments, the DEPT-135 and APT spectra of **7** (SI: S42, SI: S43) showed the absence of one methylene carbon (attributed to C-α) present in papaverine and the appearance of a new methine carbon at δ_C_72.5 ppm. In addition, seven other methine carbons (C-3, C-4, C-5, C-8, C-6′, C-5′, C-2′) and four methyl carbons (C-3′, C-4′, C-6, C-7) were present. The IR spectrum (SI: S45) showed a strong absorption band at 3437 cm^−1^, suggesting the presence of hydroxy functionality, which is lacking in the papaverine spectrum, as formerly reported. Based on **7**’s spectral data, it was identified as papaverinol N-oxide that was previously isolated through microbial transformation of papaveraldine via Mucor ramannianus 1839 [10]; however, this is the first report of papaverinol N-oxide being isolated from biotransformation of papaverine (Figure 1).

Most of the biotransformation reactions were observed to take place in the heterocyclic ring, reinforcing the binding of the aromatic ring to the active site of microbial enzymes and bringing the heterocyclic ring to direct contact with the modification site. Furthermore, the oxidative transformation of papaverine via hydroxylation, N-oxidation and dealkylation can be proposed to be mediated with cytochrome P450 monoxygenases [23].

### 2.2. ADMET/Pharmacokinetic Properties and Drug-Likeness Predictions of Papaverine and Its Metabolites *(**1**–**7**)*

Through applying the necessary procedures in drug design, development and discovery endeavors, it is crucial to analyze several vital pharmacokinetic parameters, or ADMET qualities (absorption, distribution, metabolism, excretion and toxicity), as the most relevant attributes [24,25,26].

The physicochemical characteristics of papaverine and its metabolites (**1**–**7**) are discussed in Table 3 and Figure 2A.

As shown in Table 3, the papaverine and its metabolites met the necessities of drug likeness and passed their filters, such as the Veber filter (rotatable bonds ≤ 10 with TPSA ≤ 140) [27,28]. Furthermore, the papaverine and its metabolites were also verified with Lipinski’s rule of five (MW ≤ 500, HBA ≤ 10, HBD ≤ 5 and Log Po/w ≤ 5) [29].

As shown in Figure 2B–I, the papaverine and its metabolites (**1**–**7**) were investigated and divided into six sections with adequate ranges for oral bioavailability. These characteristics included lipophilicity, insolubility, size, insaturation, polarity and flexibility [30]. The papaverine and its metabolites’ results were within these limits and had good physiochemical profiles, which is one of the factors that has to be observed in pharmaceuticals and clinical investigations (metabolites **2**–**4** were exceptions because of insaturation).

Before a biomolecule is used in pharmaceutical or clinical trial fields for pharmaceutical formulation, it is crucial to have its HIA and CNS absorption evaluated [31]. Blood–brain barrier penetration is crucial because it ensures that only substances that act on the central nervous system (CNS) can pass through it and that substances that have no effect on the CNS should not interact [32]. The BOILED- Egg curve can be used to forecast drugs’ GI absorption (HIA) and BBB penetration [33]. There are two areas: one for BBB penetration (yolk) and another for the GI absorption zone (HIA). Any component identified in the gray zone is not indicative of GI absorption or BBB penetration. The papaverine and all of its metabolites (apart from metabolite **2**) showed high gastrointestinal absorption (HIA) and high BBB permeability, as shown in Table 3. However, papaverine and other BBB-penetrating metabolites (apart from metabolite **1**) are vulnerable to P-efflux gp’s mechanism.

In skin penetration according to previous reported research [34], the log K_p_ values of papaverine and its metabolites were acceptable. The less permeant a molecule is in the skin, the bigger the negative value of K_p_. Another advantage is that it anticipates the five main cytochrome (CYP) isoforms. These enzyme isoforms handle the metabolisms of nearly 75% of the drugs sold today and play a critical role in excretion of pharmaceuticals. Inhibiting any of these isoforms results in significant drug–drug interactions [35]. Metabolites **2** and **7** did not block any cytochrome isoform (except CYP1A2), as shown in Table 4, and were rapidly metabolized. Drug–medication interactions may occur when three or four cytochrome isoforms are inhibited by remaining metabolites. Drug clearance, which is calculated via summing the excretion rates from the liver and the kidney, determines the dosage rates needed to achieve steady-state concentrations. Furthermore, the clearance value for metabolite **7** was adequate. The unfavorable interactions that arise when organic cation transporter 2 (OCT2) inhibitors and substrates are combined may be influenced with OCT2 intermediates. It has been proposed that papaverine and all of its metabolites would serve as OCT2 nonsubstrates.

Before a medication enters the clinical trial stage or the production phase of the pharmaceutical industry, it is crucial to assess its toxicological profile [36]. Each compound in this study had its many toxicities, including those that could harm the environment and people, evaluated (Table 5). The mutagenic potential of a substance can be determined using the Ames test. Results showed that papaverine and all its metabolites were classed as non-Ames-hazardous, i.e., doubtful to be carcinogenic. It is likely that blocking the potassium channels that hERG encodes would result in dangerous ventricular arrhythmia. All examined compounds can suppress hERG II but not hERG I, according to numerous studies [37]. However, it was anticipated that papaverine and metabolites **3**–**5** would be hepatotoxic and cause liver damage as a result of the drugs. It was projected that metabolites **1**–**2** and **6**–**7** would not be hepatotoxic. Dermally applied items may produce skin hypersensitivity, although none of the compounds have been proven to make people’s skin more sensitive.

Additionally, compound metabolites **1** and **2** each received a score greater than 300 mg/kg for prediction of lethal doses (LD50) and were classified as class IV; as a result, they are considered “harmful if swallowed” (300 LD50 2000). Meanwhile, papaverine and metabolites **3**–**7** had scores of over 2000 and were classified as class V; as a result, they are considered “may be harmful if swallowed” (2000) (Table 5). According to their toxicological characteristics, papaverine and all of its metabolites are categorized as being in classes IV and V because they are not thought to pose a concern of protein poisoning.

### 2.3. Docking Studies of Metabolites on the Active Sites of PTP1B (1G7F) and α-Glucosidase (3A4A)

In this study, we carried out a molecular docking study on the isolates through a molecular operating environment (MOE) program to verify that they have effects on reducing blood sugar in diabetic patients. We targeted two enzymes that are directly linked to diabetes: protein tyrosine phosphatase 1B (PTP1B) (PDB ID: 1G7F) [12,38,39] and α-glucosidase (PDB ID: 3A4A) [40,41,42]. This method was validated with redocking of 2-{4-[(2S)-2-[({[(1S)-1-carboxy-2-phenylethyl]amino}carbonyl)amino]-3-oxo-3-(pentylamino)propyl]phenoxy}malonic acid (INZ), the cocrystalized inhibitor for PTP1B (PDB: 1G7F), and alpha-D-glucopyranose (GLC), the cocrystalized inhibitor for α-glucosidase (PDB: 3A4A). In both cases, the RMSD values between the cocrystallized pose and the docked pose of the same ligand were determined. As indicated in Table 6, all the isolates (papaverine and its metabolites, **1**–**7**) demonstrated effective binding, with BE (binding energy) values that had negative kcal.mol^−1^ for both PTP1B and α-glucosidase. Among all the isolates, we found that metabolite **7** had the best docking score on both targets, with BE values of −6.86 kcal. mol^−1^ (for PTP1B) and −7.77 kcal. mol^−1^ (for α-glucosidase), as shown in Table 6.

The 2D and 3D interactions of metabolite **7** with PTP1B (BE; −6.86 kcal.mol^−1^) (Table 6 and Figure 3) revealed that the heterocyclic N-oxide moiety formed three hydrogen bond interactions with Cys215, Gly220 and Arg221, as well as one hydrogen bond between one methoxy group and Lys120. Moreover, both the isoquinoline and 3,4-dimethoxyphenyl rings formed hydrophobic interactions and π-π stacked interactions.

In addition, the 2D and 3D interaction patterns of metabolite **7** with glucosidase (BE; −7.77 kcal.mol^−1^) (Table 6 and Figure 4) revealed that one of the methoxy groups formed a hydrogen bond interaction with Gln353. Moreover, the heterocyclic N-oxide moiety formed three electrostatic interactions with Asp352, Glu411 and Arg442. Additionally, each aromatic ring and methoxy group formed hydrophobic interactions and one π-π stacked interaction.

Finally, when compared to the reference compound “acarbose”, metabolite **7** was found to display a predicted promising effect on diabetes through its action with the glucosidase target more than the phosphatase target. The docking studies of the remaining metabolites on the active sites of PTP1B (1G7F) and α-glucosidase (3A4A) are illustrated in the Appendix A.

### 2.4. Docking Studies of Metabolites on the Active Site of Lipase (PDB: 1LPB)

Here, we carried out a molecular docking study on the isolates to verify that they have an effect on reducing blood sugar in diabetic patients indirectly via acting on the lipase enzyme (PDB ID: 1LPB) [43,44,45,46]. This method was validated via redocking of octyl beta-D-glucopyranoside (BOG), the cocrystalized inhibitor (PDB: 1LPB). In addition, the RMSD value was determined. As indicated in Table 7, all the isolates (papaverine and its metabolites, **1**–**7**) demonstrated efficient binding, with BE (binding energy) values that had negative kcal.mol^−1^. Among of them, metabolite **7** also had the best docking score, with a BE value of -6.05 kcal.mol^−1^, as shown in Table 7. As described in Table 7 and Figure 5, the docking pose of metabolite **7** into the lipase active site showed electrostatic interaction between N-oxide and His30, along with several hydrophobic interactions between the methoxy groups and aromatic rings and six amino acids (Pro4, Leu16, Leu18, His30, Leu34 and Leu36).

After all, metabolite **7** was found to possess a predicted favorable effect on the lipase target more than did the reference compound “orlistat”. The docking studies of the remaining metabolites on the active site of lipase (PDB: 1LPB) are illustrated in the supplementary files (SI: S60, S61, S62, S63, S64, S65, S66).

### 2.5. MD Simulation

A 100 ns MD simulation was used to confirm the stability of docked metabolite **7** in the active sites of the three targets of interest. To validate this goal, the RMSD for the backbone was calculated and compared to the apoprotein over the entire simulation time. As shown in Figure 6, there was no obvious fluctuation in the RMSD of the docked metabolite, indicating stability of the formed complexes throughout the entire MD simulation in all studied targets. A detailed analysis of MD simulation of the lipase enzyme showed that the lipase enzyme alone demonstrated great stability with RMSD deviation from an initial conformation of less than 1 Å. Similarly, metabolite **7** showed similar stability in the complex, which indicates that the system is well balanced. Furthermore, RMSD deviation from the initial conformation of the complex was less than 2 Å. However, the complex showed higher RMSD between the 30 and 75 ns intervals, which may be attributed to the high flexibility of the mobile lid, especially in the presence of bulky substrates (ESI2) [47].

In addition, the radius of gyration of the apoproteins showed no increase in size after complexation with metabolite **7**, indicating preservation of the initial folding, as seen in Figure 7. Furthermore, simulations of solvent-accessible surface areas (SASAs) for the three enzymes were compared with their patterns after complexation with the ligands of interest and found to have nearly the same initial conformations (Figure 8).

Additionally, the RMSF was calculated via taking the average of all atoms in each residue and demonstrating the fluctuations of each residue in the three enzymes over 100 ns. The lipase and glucosidase enzymes’ amino acid residues demonstrated nearly similar fluctuation displays in the presence and absence of metabolite **7**, while in the phosphatase enzyme, some amino acid residues showed greater stabilization in the presence of metabolite **7** (Figure 9). In the tyrosine phosphatase enzyme, PRO180, ASP181 and GLY183 showed less fluctuation. Furthermore, the RMSF of the peripheral amino acid residues of the lipase enzyme showed a highly mobile lid [47], while metabolite **7** was adsorbed to the hydrophobic interface of the enzyme with good stability in this site (ESI 2). In addition, analyzing the interactions of metabolite **7** with the three studied enzymes over the whole simulation time revealed different types of interaction, which was in good agreement with the docking results (Figure 10). From all the above results, the complexes of metabolite **7** with the three enzymes showed good stability in the active site of each enzyme.

## 3. Materials and Methods

### 3.1. General Experimental Procedure

The 1D (^1^H, ^13^C, APT and DEPT 135) and 2D (HMBC, COSY and HSQC) NMR spectra were acquired using standard pulse sequences on a Bruker model AMX 400 NMR spectrometer operating at 400 MHz and 100 MHz in ^1^H NMR and ^13^C NMR, respectively. Results for the chemical shift (δ) were provided in parts per million units (ppm). J values for coupling constants were provided in Hertz (Hz). The HSQC, HMBC, APT and DEPT pulse sequences followed technical standards. For EI-MS, mass spectroscopy with a Thermo Scientific ISQ single-quadrupole mass spectrometer was used. For ESI-MS, a mass spectrometer from Thermo Scientific with a triple-quadrupole Access MAX system and Xcalibur 2.1 software was used. Sephadex LH-20 and silica gel (E. Merck, 70–230 mesh) column chromatography were used for obtaining pure metabolites. Analytical-grade chemicals were used in isolation and purification. Merck TLC sheets and silica gel G 254 F sheets were used for TLC (E. Merck, Germany). Solvent systems S1, ethyl acetate–methanol–ammonia (95:5:5), and S2, EtAc-MeOH-H_2_O (4:1:5, *v*/*v*), were utilized for TLC analysis. Dragendorff’s and anisaldehyde/sulfuric acid spray reagents and a UV lamp were used to visualize the TLC plates (at 254 and 365 nm). Papaverine was obtained from Sigma-Aldrich (St. Louis, MO, USA).

### 3.2. Microorganism

Preliminary detection of metabolite formation was carried out as previously reported [7]. Fifty microbial cultures were obtained from the Northern Regional Research Laboratories in Peoria, Illinois and the American Type Culture Collection in Rockville, Maryland, both in the United States. The lyophilized microorganisms were used to initiate the cultures. *Penicillium chrysogeneum* ATCC 10002, *Cunninghamella elegans* NRRL 2310, *Rhodotorula rubra* NRRL y1592 and *Cunninghamella blackesleeana* NRRL 1369 were the most efficient microorganisms at converting papaverine into metabolites without optimization. Using the same method, substantial amounts of metabolites were produced. The highest-yielding microbial strains that could biotransform papaverine were used in the preparative-scale fermentation. All fermentation studies used a sterilized liquid medium, at 121 °C for 15 min, containing the following constituents: one liter of distilled water, 5 gm of NaCl, 5 gm of K_2_HPO_4_, 5 gm of peptone and 5 gm of yeast extract. The fermentations were discontinued once conversion reached its maximum level, then filtered over cheesecloth and basified via adding one mL of NH_3_/30 mL of culture broth (pH 8). The fermentation broths were then extracted until exhaustion with an equal volume of ethyl acetate, dehydrated over anhydrous Na_2_SO_4_ and evaporated to dryness under vacuum using a rotary evaporator to provide fermentation residues.

### 3.3. Large-Scale Fermentation and Isolation of the Metabolites

#### 3.3.1. Papaverine Transformation via Cunninghamella Elegans NRRL 2310

A solution of papaverine in DMF (800 mg/20 mL) was distributed among 40 flasks (500 mL each), each containing 100 mL of a stage II culture. After 10 days, 1.5 g of fermentation residue was obtained. Two major spots were detected on prespread TLC sheets of silica gel, with R_f_ values = 0.5 and 0.17 (solvent system S1) and 0.43 and 0.2 (solvent system S2), respectively. The residue was loaded on top of a silica gel column (100 g, 120 cm × 2.5 cm) after being dissolved in a 1:1 methanol-dichloromethane combination (3 mL). A gradient elution method was adopted, starting with methylene chloride, then moving to the methylene chloride–ethyl acetate mixture with gradually increasing polarity via added ethyl acetate in 5% increments of up to 100% ethyl acetate, then moving to the ethyl acetate–methanol mixture with gradually increasing polarity via added methanol in 10% increments of up to 70% methanol. One hundred and fifty fractions were collected (10 mL each). Fractions (33–45) yielded pure **1** as a white, amorphous powder (18 mg, 2.3% yield, R_f_ 0.5, S1). Fractions (131–139) yielded pure **2** as a white, amorphous powder (17 mg, 2.1% yield, R_f_ 0.17, S1).

#### 3.3.2. Papaverine Transformation via Rhodotorula Rubra NRRL y1592

In this preparative scale fermentation, 40 flasks (500 mL each), each containing 100 mL of a stage II culture, were used. A solution of papaverine in DMF (400 mg/10 mL) was distributed amongst these flasks. In total, 850 mg of fermentation residue was recovered after 12 days. Using the S1 solvent system, two spots were located on Merck TLC sheets of silica gel (R_f_ = 0.9, 0.64). The residue was loaded on top of a silica gel column (90 g, 120 cm × 2.5 cm) after being solubilized in 2 mL of a MeOH-CH_2_Cl_2_ mixture (1:1). The ethyl acetate was used as the mobile phase for the gradient elution procedure, and the polarity was then increased with methanol in 5% increments of up to 80% methanol; 25 mL fractions were collected. Fractions (40–66) and (70–85) were pooled and evaporated to produce 20 mg and 25 mg of impure 3 and impure 4 residue, respectively. Column chromatography on Sephadex LH-20 (5 g, 60 cm × 1 cm) was used to further purify these compounds via elution with 100% methanol. Thirty fractions, each containing 5 mL, were collected. Pure compound **3** was produced via combining and evaporating fractions 15–20. This produced a white, amorphous powder, with R_f_ = 0.9 in S1, that was soluble in chloroform (5 mg, yield: 1.25%). For additional purification of **4**, the same Sephadex LH-20 column (5 g, 60 cm × 1 cm) and 100% methanol elution were used. Twenty-five fractions, each containing 5 mL, were collected. Fractions (16–22) were combined and evaporated to give pure compound **4** as a white, amorphous powder that was soluble in chloroform (6 mg, 1.5% yield, R_f_ 0.64, S1).

#### 3.3.3. Papaverine Transformation via Cunninghamella Blackesleeana NRRL 1369

Forty flasks of similar size (500 mL), each containing a stage II culture (100 mL), each received 400 mg of papaverine in 20 mL of DMF. In total, 800 mg of fermentation residue was recovered after 15 days of fermentation. The residue was loaded on top of a silica gel column (90 g, 120 cm × 2.5 cm) after being dissolved in a CH_3_OH-CH_2_Cl_2_ mixture (1:1) (2 mL). Ethyl acetate was used as the mobile phase for the gradient elution procedure. The polarity was gradually increased via adding methanol in increments of 2%, of up to 80% methanol; 25 mL fractions were collected. Fractions (30–36) were collected and evaporated to produce impure **5** residue. Further purification was carried out using the Sephadex LH-20 column (5 g, 60 cm × 1 cm), which was eluted with 100% methanol. White, amorphous powder made up pure compound **5**, which had an R_f_ value of 0.35 using S1. Metabolite **5** was soluble in chloroform (6 mg, 1.5% yield).

#### 3.3.4. Papaverine Transformation via Penicillium Chrysogeneum ATCC 10002

In this fermentation, 40 flasks (each 500 mL), each containing 100 mL of a stage II culture, were fed with equal amounts of papaverine solution (10 mg/0.25 mL DMF per flask). In total, 1.3 g of fermentation residue was obtained after two weeks. Using the S1 solvent system, two major spots were detected on precoated TLC sheets of silica gel (R_f_ = 0.15, 0.10). The residue was solubilized in 2.5 mL of methanol–dichloromethane (1:1) before being loaded onto a silica gel column (110 g, 120 cm × 2.5 cm). The ethyl acetate was initially used for the gradient elution procedure. Polarity was steadily increased via addition of methanol in 5% increments of up to 90% methanol; 30 mL fractions were collected. Combining and evaporating fractions 60–66 produced 15 mg of impure **6** residue, while mixing and evaporating fractions 75–82 produced 10 mg of impure **7** residue.

The use of a Sephadex LH-20 column (5 g, 60 cm × 1 cm) and elution with 100% methanol was adopted to further purify **6**. A total of 40 fractions (5 mL each) were collected. Mixing and evaporating fractions (28–35) produced pure compound **6** as a white, amorphous powder soluble in chloroform (6 mg, 1.5% yield, R_f_ = 0.15, S1). The Sephadex LH-20 column (5 g, 60 cm × 1 cm) was also used to obtain pure **7** via repeated elution with 100% methanol. Thirty fractions, 5 mL each, were collected. Fractions (20–25) were combined and evaporated to produce pure compound **7** as a white, amorphous powder (1.5% yield, R_f_ = 0.10, S1).

### 3.4. ADMET/Pharmacokinetic Properties and Drug-Likeness Predictions

Before a compound is chosen as a therapeutic candidate, ADMET (absorption, distribution, metabolism, excretion and toxicity) measurements are crucial. The ADMET properties were obtained using the online web tool Swiss ADME (http://www.swissadme.ch/index.php, accessed on 20 January 2023), and the pharmacokinetic scores were predicted using the online web application pkCSM (http://biosig.unimelb.edu.au/pkcsm/prediction, accessed on 20 January 2023).

### 3.5. Molecular Docking Studies

The binding manners and interactions of the isolates (papaverine and its metabolites **1**–**7**) were demonstrated using molecular docking studies (MOE version 2020.9010) and the Discovery Studio (DS) visualizer tool. The three crystal structures of the human PTP1B catalytic domain, α-glucosidase and lipase, with resolutions of 1.80 (PDB ID: 1G7F), 1.60 (PDB ID: 3A4A) and 2.46 (PDB ID: 1LPB), respectively, were obtained from the Protein Data Bank. The compounds were predocked with their ligands. There were hydrophilic and hydrophobic amino acids present in each enzyme’s ligand-binding site. A redocking approach for each ligand was used to confirm the docking process through establishing a number of docked poses, one of which had an RMSD value less than 1 (0.85 (with INZ), 0.94 (with GLC) and 0.80 (with BOG)). All of the isolates examined fit perfectly into the active pocket of the enzyme, according to the results of the molecular docking study. Furthermore, the most favorable docked poses of each isolate were examined for binding-mode analysis based on the findings of the binding free energy calculation.

### 3.6. MD Simulation

Molecular dynamics (MD) simulation studies were performed with GROMACS 2021 on the best docking poses of metabolite **7** [48]. The ligand topology was generated using the CHARMM General Force Field (CGenFF) server, while the CHARMM forcefield parameters were applied to the studied proteins [49]. A cubic box of the TIP3P water model with 10 padding was utilized, and the system was neutralized through addition of Na^+^ and Cl^-^ ns. The CHARMM36 forcefield was applied on the protein–ligand complex [50]. A cutoff distance of 12 A was applied for nonbonded interactions and long-range electrostatic interactions. The Verlet cutoff-scheme [51] and the particle-mesh Ewald (PME) method [52] were used for searching the neighbor list and the long-range electrostatic interactions, respectively. The simulation parameters were set as in the literature [53], except for the production time, where 100 ns was applied. GROMACS utilities were used for the analysis of the MD simulations [54]. Trajectory visualization and analysis were performed on a VMD molecular graphics program.

## 4. Conclusions

In conclusion, seven metabolites were isolated from the fungal bioconversion of papaverine: namely, 3,4-dihydropapaverine (metabolite **1**), papaveroline (metabolite **2**), 7-demethyl papaverine (metabolite **3**), 6,4′-didemethyl papaverine (metabolite **4**), papaverine-3-ol (metabolite **5**), papaverinol (metabolite **6**) and papaverinol N-oxide (metabolite **7**). This biotransformation included reduction, demethylation, N-oxide formation, oxidation and hydroxylation reactions. In silico molecular docking and molecular dynamics simulation studies against DM were used to examine the efficacy of safe bioactive compounds from filamentous fungi in preventing the actions of certain enzymes. Important enzymes linked to DM illness include PTP1B, α-glucosidase and PL, and their activity can be targeted and suppressed. Based on its superior binding affinities to these three targets and its favorable safety profile, metabolite **7** (papaverinol N-oxide) was chosen following a thorough molecular docking investigation of bioactive compounds. Additionally, analysis of molecular dynamics simulation at 100 ns supported each target–ligand complex of metabolite **7**’s stability. It is necessary to conduct more in vitro and in vivo animal research to assess this bioactive metabolite’s aphrodisiac effects.

## Figures and Tables

**Figure 1 molecules-28-01583-f001:**
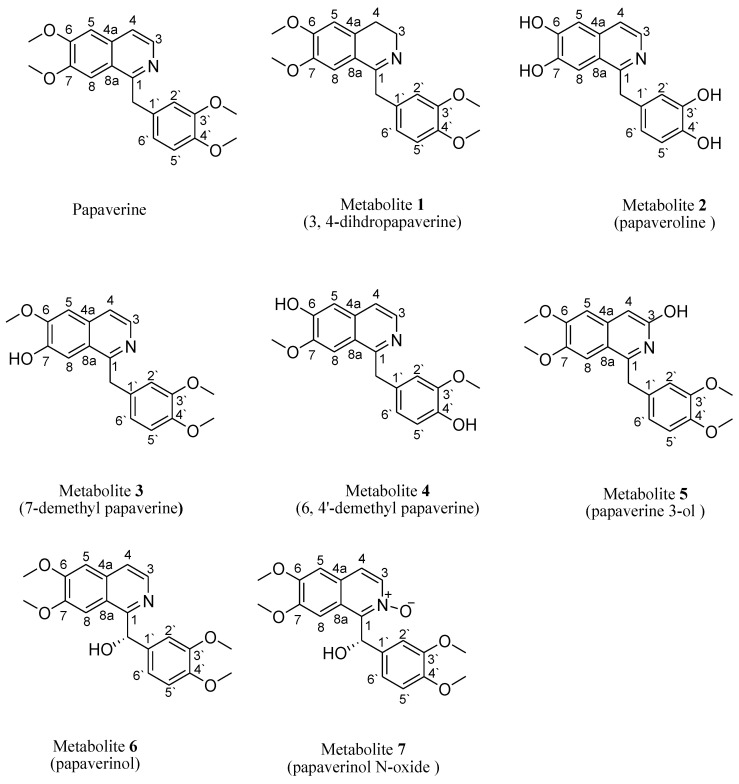
Structure of papaverine and its metabolites.

**Figure 2 molecules-28-01583-f002:**
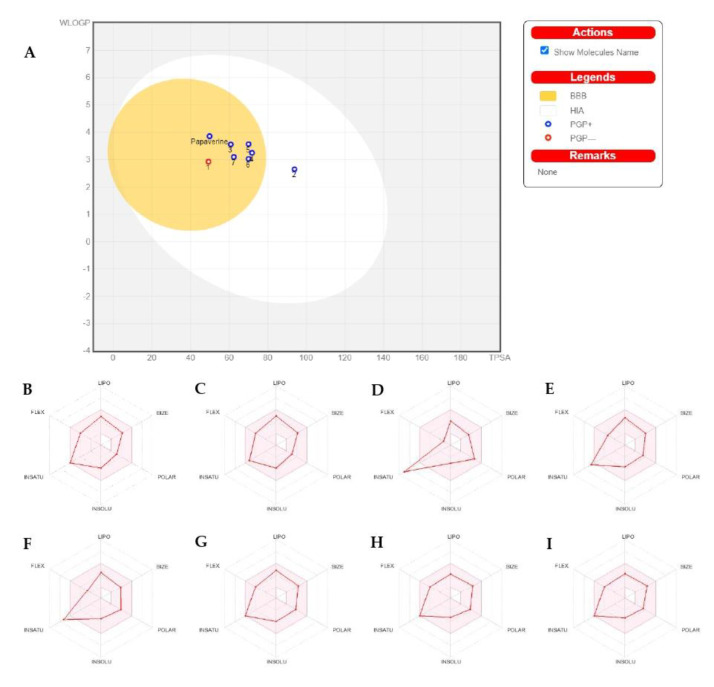
The BOILED-Egg diagram for papaverine and all its metabolites (**A**), and bioavailability radar charts for papaverine and each metabolite (**B**–**I**): (**B**) for papaverine, (**C**) for metabolite **1**, (**D**) for metabolite **2**, (**E**) for metabolite **3**, (**F**) for metabolite **4**, (**G**) for metabolite **5**, (**H**) for metabolite **6** and (**I**) for metabolite **7**.

**Figure 3 molecules-28-01583-f003:**
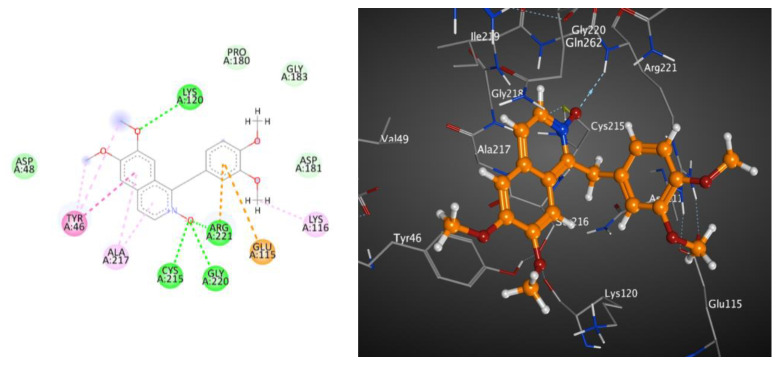
These 2D (**left**) and 3D (**right**) patterns demonstrate the binding interaction of metabolite **7** into the active site of PTP1B (PDB ID: 1G7F).

**Figure 4 molecules-28-01583-f004:**
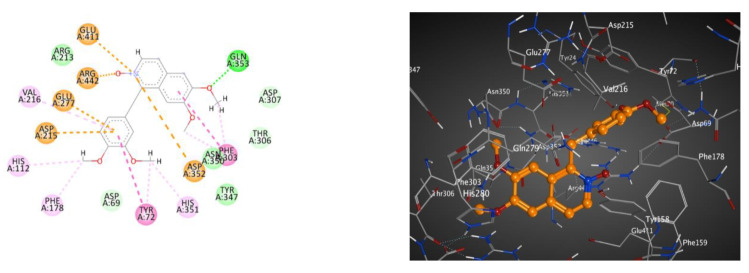
These 2D (**left**) and 3D (**right**) patterns demonstrate the binding interaction of metabolite **7** into the active site of glucosidase (PDB ID: 3A4A).

**Figure 5 molecules-28-01583-f005:**
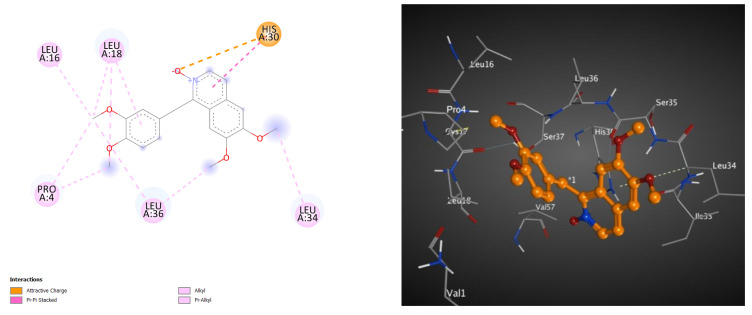
These 2D (**left**) and 3D (**right**) patterns demonstrate the binding interaction of metabolite **7** into the active site of lipase (PDB ID: 1LPB).

**Figure 6 molecules-28-01583-f006:**
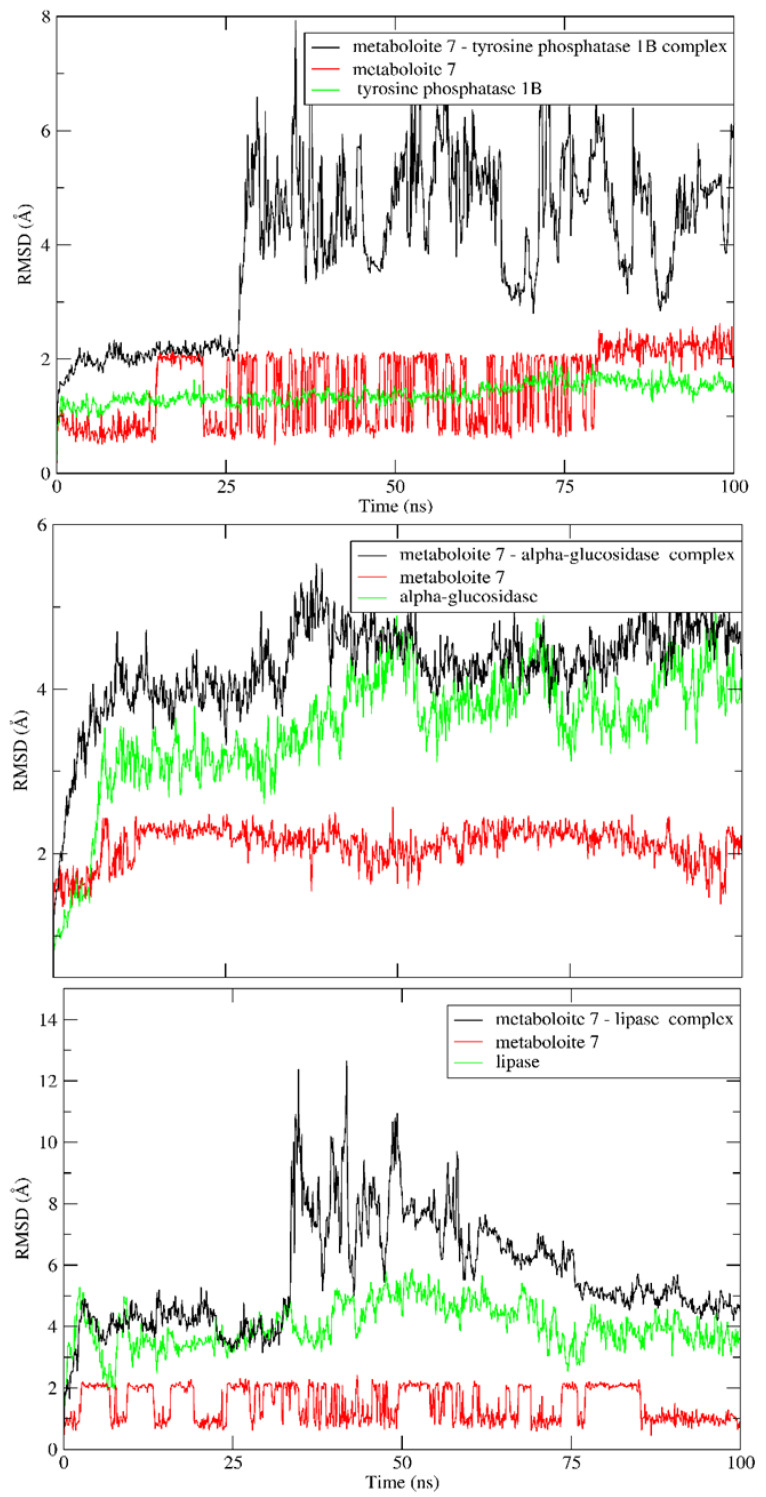
RMSD values of apoprotein of metabolite **7** and its complexes with the tyrosine phosphatase 1B, α- glucosidase and lipase enzymes over a 100 ns MD simulation.

**Figure 7 molecules-28-01583-f007:**
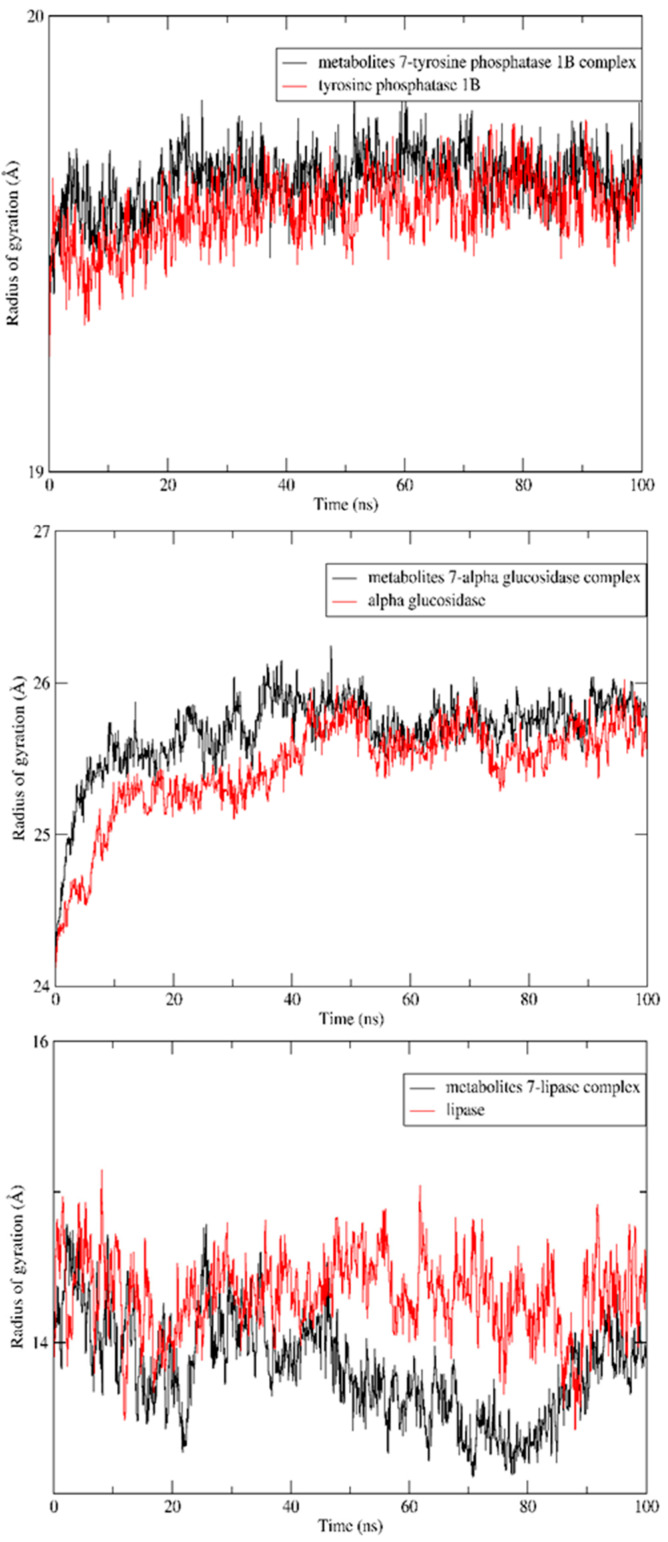
Radiuses of gyration of apoprotein of the tyrosine phosphatase 1B, α- glucosidase and lipase enzymes and its complex with metabolite **7** over a 100 ns MD simulation.

**Figure 8 molecules-28-01583-f008:**
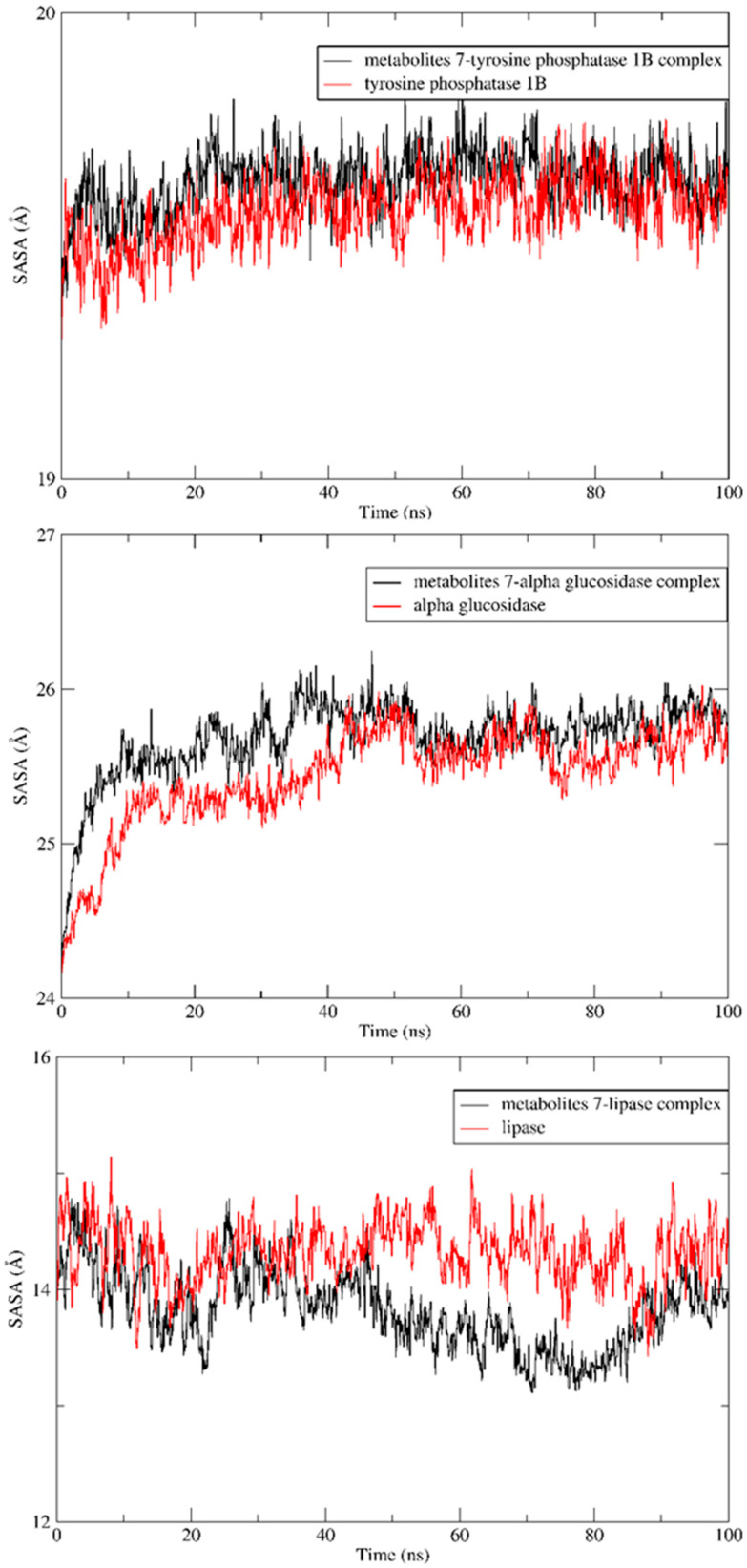
SASA analyses for both LdMetAP-1 and LdMetAP-2 over a 100 ns MD simulation.

**Figure 9 molecules-28-01583-f009:**
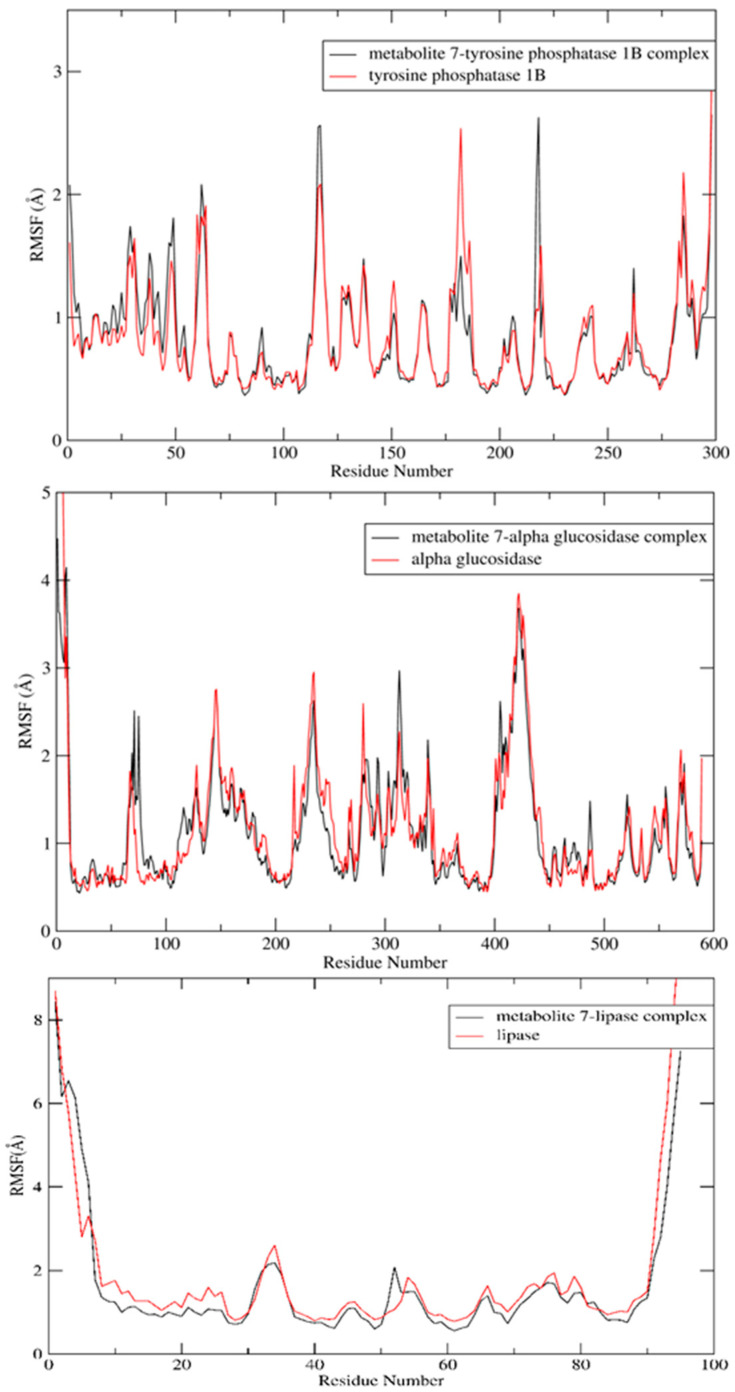
RMSF values of apoprotein of metabolite **7** and its complexes with the tyrosine phosphatase 1B, α-glucosidase and lipase enzymes over a 100 ns MD simulation.

**Figure 10 molecules-28-01583-f010:**
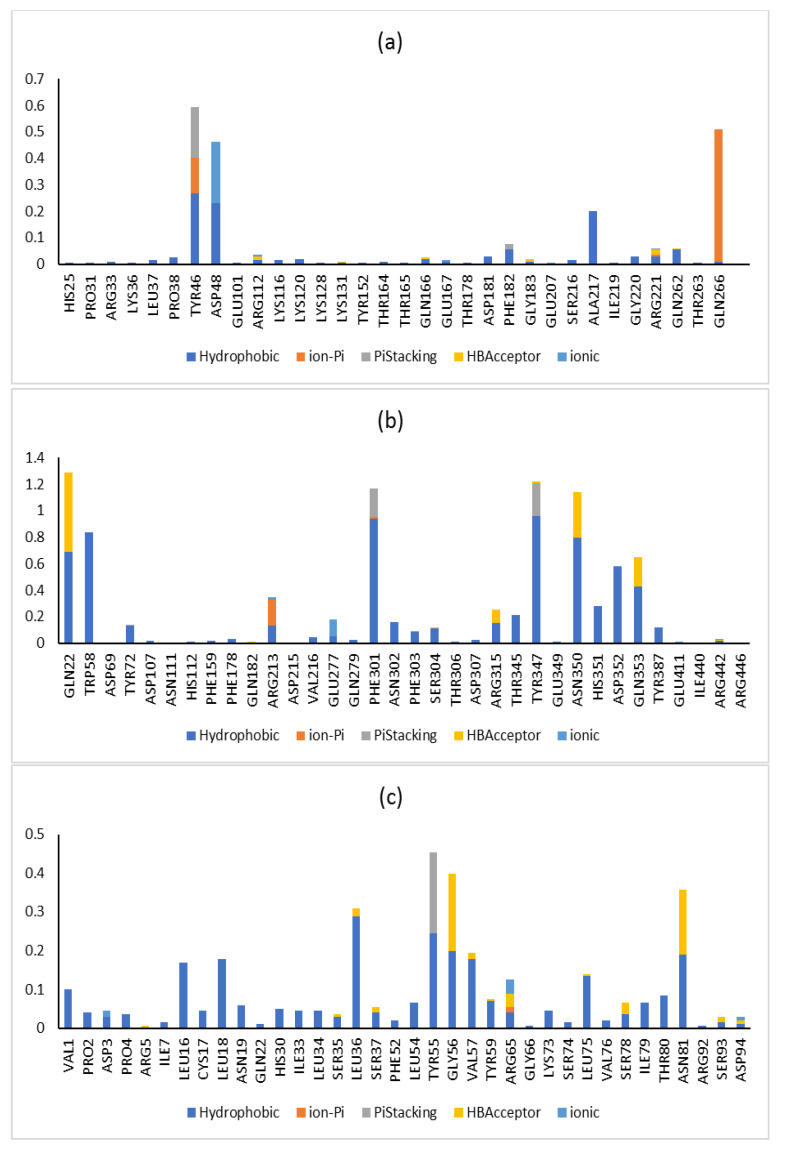
Protein–ligand contact histogram for metabolite **7** and its (**a**) tyrosine kinase 1B, (**b**) α-glucosidase and (**c**) lipase complexes.

**Table 1 molecules-28-01583-t001:** Data from ^1^H NMR spectroscopy at 400 MHz for papaverine and its metabolites, **1** through **7** (in CDCl_3_).

Position	δ_H_ (Multiplicities, *J* in Hz)							
	Papaverine	1	2	3	4	5	6	7
3	8.33, *d* (5.6)	3.47, *t* (7.2)	8.31, *d* (5.7)	8.29, *d* (5.2)	8.36, *d* (5.6)	-	8.35, *d* (5.6)	8.41, *d* (5.2)
4	7.53, *d* (5.7)	3.07, *t* (7.2)	7.50, *d* (5.6)	7.52, *d* (5.2)	7.41, *d* (5.6)	6.99, *s*	7.52, *d* (5.6)	7.51, *d* (5.2)
5	7.01, *s*	7.06, *s*	7.14, *s*	7.07, *s*	7.34, *s*	7.05, *s*	7.02, *s*	7.06, *s*
8	7.41, *s*	7.38, *s*	7.45, *s*	7.54, *s*	7.40, *s*	7.33, *s*	7.07, *s*	7.12, *s*
2′	6.94, *d* (1.8)	6.79, *d* (1.6)	6.93, *d* (1.6)	6.82, *d* (1.2)	6.92, *d* (1.2)	6.61, *d* (1.2)	6.79, *d* (1.2)	6.82, *d* (1.6)
5′	6.74, brs	6.87, *d* (7.6)	6.77, *d* (8.4)	6.70, *d* (8.4)	6.75, *d* (8.0)	6.76, *d* (8.4)	6.75, *d* (8.4)	6.78, *d* (8.4)
6′	6.82, *m*	6.81, *dd* (7.6, 1.6)	6.82, *dd* (8.4, 1.6)	6.73, *dd* (8.4, 1.2)	6.88, *dd* (8.0, 1.2)	6.83, *dd* (8.4, 1.2)	6.89, *dd* (8.4, 1.2)	6.93, *dd* (8.4, 1.6)
α	4.56, *s*	4.54, *s*	4.45, *s*	4.45, *s*	4.55, *s*	4.43, *s*	6.08, *s*	6.15, *s*
6-Ome	4.02, *s*	3.99, *s*	-	4.02, *s*	-	3.98, *s*	3.95, *s*	4.03, *s*
7-Ome	3.93, *s*	3.90, *s*	-	-	3.91, *s*	3.88, *s*	3.71, *s*	3.90, *s*
3′-Ome	3.78, *s*	3.76, *s*	-	3.72, *s*	3.76, *s*	3.76, *s*	3.76, *s*	3.76, *s*
4′-Ome	3.80, *s*	3.81, *s*	-	3.77, *s*	-	3.82, *s*	3.81, *s*	3.82, *s*
6-OH	-	-	5.06, *s*	-	6.03, *br s*	-	-	-
7-OH	-	-	5.15, *s*	6.02, *br s*	-	-	-	-
3′-OH	-	-	5.33, *s*	-	-	-	-	-
4′-OH	-	-	5.64, *s*	-	5.94, *br s*	-	-	-
3-OH	-	-	__	-	-	2.36, *br s*	-	-
α-OH	-	-	__	-	-	-	6.38, *br s*	6.50, *br s*

**Table 2 molecules-28-01583-t002:** Data from ^13^C NMR spectroscopy at 100 MHz for papaverine and its metabolites, **1** through **7** (in CDCl_3_).

Position		δ_H_ (Multiplicities, *J*)						
	Papaverine	1	2	3	4	5	6	7
1	156.9, C	156.5, C	157.5, C	156.4, C	157.3, C	158.9, C	157.0, C	155.6, C
3	139.0, CH	46.5, CH_2_	141.7, CH	139.9, CH	141.2, CH	165.9, C	138.6, CH	138.0, CH
4	119.6, CH	26.2, CH_2_	119.0, CH	119.3, CH	117.7, CH	98.3, CH	119.5, CH	119.8, CH
4a	134.6, C	134.1, C	133.5, C	132.0, C	133.6, C	134.0, C	132.8, C	133.5, C
5	105.6, CH	104.8, CH	105.7, CH	105.3, CH	103.2, CH	108.7, CH	105.0, CH	105.2, CH
6	150.7, C	153.4, C	150.5, C	150.1, C	150.8, C	151.7, C	152.2, C	151.9, C
7	149.3, C	150.4, C	148.4, C	147.8, C	147.2, C	149.4, C	149.5, C	149.7, C
8	104.5, CH	103.3, CH	104.1, CH	102.2, CH	103.7, CH	105.1, CH	102.6, CH	103.6, CH
8a	122.9, C	121.5, C	122.1, C	122.1, C	121.9, C	118.1, C	120.9, C	120.9, C
1′	131.6, C	133.2, C	133.0, C	134.3, C	133.2, C	131.9, C	135.6, C	136.1, C
2′	111.3, CH	111.1, CH	112.0, CH	112.1, CH	111.5, CH	110.6, CH	111.3, CH	110.9, CH
3′	147.9, C	148.8, C	146.1, C	149.1, C	148.0, C	149.7, C	148.0, C	148.4, C
4′	149.3, C	149.3, C	145.0, C	148.1, C	146.2, C	146.8, C	150.2, C	149.0, C
5′	112.2, CH	109.2, CH	110.1, CH	110.4, CH	110.3, CH	112.3, CH	110.2, CH	110.3, CH
6′	120.7, CH	120.6, CH	120.8, CH	120.6, CH	120.3, CH	122.3, CH	120.0, CH	120.0, CH
α	40.6, CH_2_	47.0, CH_2_	43.4, CH_2_	44.6, CH_2_	42.4, CH_2_	42.4, CH_2_	72.5, CH	72.4, CH
6-Ome	55.9, CH_3_	55.5, CH_3_	-	56.3, CH_3_	-	56.3, CH_3_	55.9, CH_3_	55.9, CH_3_
7-Ome	56.10, CH_3_	55.6, CH_3_	-	-	55.3, CH_3_	56.1, CH_3_	55.7, CH_3_	55.7, CH_3_
3′-Ome	56.12, CH_3_	55.7, CH_3_	-	56.2, CH_3_	55.5, CH_3_	56.0, CH_3_	55.6, CH_3_	55.6, CH_3_
4′-Ome	56.4, CH_3_	55.8, CH_3_	-	55.8, CH_3_	-	55.3, CH_3_	55.6, CH_3_	55.6, CH_3_

**Table 3 molecules-28-01583-t003:** Predicted physicochemical parameters and lipophilicity properties of papaverine and its metabolites, **1**–**7**.

	Papaverine	Metabolite	Metabolite	Metabolite	Metabolite	Metabolite	Metabolite	Metabolite
		1	2	3	4	5	6	7
MW ^a^	339.39	341.40	283.28	325.36	311.33	355.39	355.39	355.39
Rot. Bond ^b^	6	6	2	5	4	6	6	6
HBA ^b^	5	5	5	5	5	6	6	5
HBD ^c^	0	0	4	1	2	1	1	0
TPSA ^d^	49.83	49.30	93.80	60.82	71.82	70.06	70.06	62.40
Log P_o/w_	3.86	3.30	2.64	3.55	3.25	3.56	3.35	3.09

^a^ Molecular weight (g/mol), ^b^ H-bond acceptor, ^c^ H-bond donor, ^d^ topological polar surface area.

**Table 4 molecules-28-01583-t004:** Predicted pharmacokinetic parameters of papaverine and its metabolites, **1**–**7**.

	Papaverine	Metabolite	Metabolite	Metabolite	Metabolite	Metabolite	Metabolite	Metabolite
		1	2	3	4	5	6	7
**Absorption**								
Water Solubility	−4.79	−4.94	−3.23	−4.47	−4.01	−4.21	−3.70	−4.17
GI	100	100	79.85	98.62	95.14	95.89	96.54	98.24
Log K_p_	−2.72	−2.72	−2.76	−2.64	−2.77	−2.73	−2.73	−2.66
**Distribution**								
BBB	0.37	0.04	−1.10	0.18	−0.47	−0.47	−0.49	0.13
Log PS	−2.41	−2.36	−2.46	−2.37	−2.39	−3.19	−3.31	−2.43
VD	0.03	0.13	−0.05	0.13	0.06	-0.13	0.03	0.64
**Metabolism**								
CYP1A2 Inhibitor	√	√	√	√	√	√	√	√
CYP2C9 Inhibitor	√	√	X	√	√	√	√	X
CYP2C19 Inhibitor	√	√	X	√	√	√	√	X
CYP3A4 Inhibitor	X	X	X	X	X	X	X	X
CYP2D6 Inhibitor	X	X	X	X	X	√	√	X
**Excretion**								
Total Clearance	0.37	0.51	0.14	0.33	0.25	0.71	0.44	1.00
Renal OCT2 Sub.	X	X	X	X	X	X	X	X

**Table 5 molecules-28-01583-t005:** Predicted toxicity profiles of papaverine and its metabolites, **1**–**7**.

	Papaverine	Metabolite	Metabolite	Metabolite	Metabolite	Metabolite	Metabolite	Metabolite
		1	2	3	4	5	6	7
Ames Toxicity	X	X	X	X	X	X	X	X
hERG I Inhibitor	X	X	X	X	X	X	X	X
hERG II Inhibitor	√	X	√	√	√	√	√	√
Oral Toxicity	2202	2000	1930	2100	2064	2074	2029	2374
Oral Toxicity Classification	V	IV	IV	V	V	V	V	V
Hepatotoxicity	√	X	X	√	√	√	X	X
Skin Sensitivity	X	X	X	X	X	X	X	X

**Table 6 molecules-28-01583-t006:** Docking results for all isolates in the active sites of PTP1B (PDB: 1G7F) and α-glucosidase (PDB: 3A4A).

Isolate	Binding Energy (BE) (kcal.mol^−1^)	
	PTP1B (PDB: 1G7F)	α-Glucosidase (PDB: 3A4A)
Papaverine	−6.18	−6.97
Metabolite **1**	−6.08	−4.80
Metabolite **2**	−5.74	−5.78
Metabolite **3**	−6.22	−5.28
Metabolite **4**	−6.01	−6.26
Metabolite **5**	−6.57	−4.86
Metabolite **6**	−6.72	−5.12
Metabolite **7**	−6.86	−7.77
Redocked Ligand	−7.78	−5.77
Acarbose	−7.90	−6.57

**Table 7 molecules-28-01583-t007:** Docking results for all isolates in the active site of lipase (PDB: 1LPB).

Isolate	Binding Energy (BE) (kcal.mol^−1^) Lipase (PDB: 1LPB)
Papaverine	−5.60
Metabolite **1**	−5.83
Metabolite **2**	−5.19
Metabolite **3**	−5.81
Metabolite **4**	−5.69
Metabolite **5**	−5.82
Metabolite **6**	−5.66
Metabolite **7**	−6.05
Redocked Ligand	−5.67
Orlistat	−6.28

## Data Availability

All data is contained within this article and its Appendix A.

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
