# Peer review of "Papaverinol-N-Oxide: A Microbial Biotransformation Product of Papaverine with Potential Antidiabetic and Antiobesity Activity Unveiled with In Silico Screening"

_molecules, 2023, doi:10.3390/molecules28041583_

Round 1
Reviewer 1 Report
Quite an interesting study by the authors.
The following are my concerns:
1- Nothing have been said about the MD simulation results in the abstract
2- Implications and summary findings need to be clearly stated at the last paragraph of the introduction
3- Authors claimed to have done MD simulations to validate their docking studies, however these facts are no where to be found within the manuscript in terms of figures presentation and tables.
4- The discussion on antidiabetic and antiobesity potentials of the ligands ware visibly scanty!!!
Author Response
Reviewer 1
Thank you for your precious time and your valuable comments
1- Nothing have been said about the MD simulation results in the abstract
Reply: MD simulations results added as required and marked up using the “Track Changes” function
2- Implications and summary findings need to be clearly stated at the last paragraph of the introduction
Reply: corrected as required and marked up using the “Track Changes” function
3- Authors claimed to have done MD simulations to validate their docking studies, however these facts are nowhere to be found within the manuscript in terms of figures presentation and tables.
Reply: MD simulations added as required and marked up using the “Track Changes” function
4- The discussion on antidiabetic and antiobesity potentials of the ligands ware visibly scanty!!!
Reply: corrected as required and marked up using the “Track Changes” function

Reviewer 2 Report
The authors described the biotransformation of natural heterocyclic alkaloid papaverine by filamentous fungi and isolated 7 compounds. The chemistry isolation work is great but pharmacological work has some flaws. The authors used In-silico Molecular docking technique to prove that compounds have pharmacological activity, well beyond wet lab data only the base of docking we cannot prove that any compound is pharmacologically active against any disease. it needs proper proof. Kindly check my comments and correct your article according to suggestions.
1. In abstract; Line 24, authors mentioned molecular docking simulations, while the authors didn't perform MD simulations so kindly correct this mistake.
2. The authors only mentioned compound 7 docking results while other are missing why? the author should provide it in paper or in supplementary.
3. why not the authors have done MD simulations which further strengthen the molecular docking results.
4. I strongly recommend the authors at least do some more in-silico work like ADMET otherwise just on the basis of Molecular docking it is totally unacceptable to claim any compound for any specific disease. The authors can follow this article. Molecules 2022, 27(3), 917;
5. For docking results the authors should also use some control to compare the results, only docking scores are not enough.
6. conclusions are missing.
7. Discussion portion is not well discussed.
Author Response
Reviewer 2
Thank you for your precious time and your valuable comments
1-In abstract; Line 24, authors mentioned molecular docking simulations, while the authors didn't perform MD simulations so kindly correct this mistake.
Reply: MD simulations added as required and marked up using the “Track Changes” function
- The authors only mentioned compound 7 docking results while other are missing why? the author should provide it in paper or in supplementary.
Reply: added as required, marked up using the “Track Changes” function and added to the supplementary file : S46, S47, S48, S49, S50, S51, S52, S53, S54, S55, S56, S57, S58, S59, S60, S61, S62, S63, S64, S65, S66
- why not the authors have done MD simulations which further strengthen the molecular docking results.
Reply: MD simulations added as required and marked up using the “Track Changes” function
- I strongly recommend the authors at least do some more in-silico work like ADMET otherwise just on the basis of Molecular docking it is totally unacceptable to claim any compound for any specific disease. The authors can follow this article. Molecules 2022, 27(3), 917;
Reply: ADMET added as required and marked up using the “Track Changes” function
- For docking results the authors should also use some control to compare the results, only docking scores are not enough.
Reply: we performed docking results for the positive control acarbose in the active sites of PTP1B (PDB: 1G7F) and α-glucosidase (PDB: 3A4A), according to the reference Martin, A. E., & Montgomery, P. A. (1996). Acarbose: an alpha-glucosidase inhibitor. American journal of health-system pharmacy: AJHP : official journal of the American Society of Health-System Pharmacists, 53(19), 2277–2337. https://doi.org/10.1093/ajhp/53.19.2277 (marked with highlight)
Also, the positive control Orlistat in the active site of lipase (PDB: 1LPB) according to the reference Nurul Islam, M., Jung, H. A., Sohn, H. S., Kim, H. M., & Choi, J. S. (2013). Potent α-glucosidase and protein tyrosine phosphatase 1B inhibitors from Artemisia capillaris. Archives of pharmacal research, 36(5), 542–552. https://doi.org/10.1007/s12272-013-0069-7(marked with highlight)
- Conclusions are missing.
Reply: corrected as required and marked up using the “Track Changes” function
- Discussion portion is not well discussed.
Reply: corrected as required and marked up using the “Track Changes” function

Round 2
Reviewer 1 Report
Authors should provide the contribution of each amino acids residues and interactions after MD simulation in form of bar charts.
Author Response
Reviewer 1
Thank you for your precious time and your valuable comments
Authors should provide the contribution of each amino acids residues and interactions after MD simulation in form of bar charts.
Answer: Added in page 17 with highlight: In addition, analyzing the interaction of metabolite 7 with the three studied enzymes over the whole simulation time revealed different types of interaction which was in good agreement with the docking results (Figure 10)., Also Figure 10. protein-ligand contact histogram for metabolite 7 and (a)Tyrosine kinase 1B, (b) α-glucosidase, and (c) lipase complexes.

Reviewer 2 Report
The authors addressed all my questions very well. I recommend for its publication.
Author Response
Reviewer 2
Thank you for your precious time and your valuable comments

Round 3
Reviewer 1 Report
No further comment.
Author Response
Reviewer 1
Thank you for your precious time and your valuable comments
